# Neutralization of Marinobufagenin Demonstrates Efficacy In Vitro and In Vivo in Models of Pre-Eclampsia

**DOI:** 10.3390/biomedicines13040782

**Published:** 2025-03-24

**Authors:** Ahmed F. Pantho, Mehruba Zaman, Syeda H. Afroze, John M. Wages, Bo Yu, James W. Larrick, Thomas J. Kuehl, Niraj Vora, Mohammad Nasir Uddin

**Affiliations:** 1Orion Institute for Translational Medicine, Temple, TX 76504, USA; ahmed.pantho@artemisbiotech.org (A.F.P.); afroze751@gmail.com (S.H.A.); 2Artemis Biotechnologies LLC, Temple, TX 76502, USA; tjkuehl@peoplepc.com; 3Virginia Commonwealth University School of Medicine, Richmond, VA 23298, USA; zamanm3@vcu.edu; 4Panorama Research, Inc., 1230 Bordeaux Dr., Sunnyvale, CA 94089, USA; wagesjm@gmail.com (J.M.W.); boyu9@yahoo.com (B.Y.); jwlarrick@gmail.com (J.W.L.); 5Baylor Scott & White Health Temple Medical Center, Temple, TX 76508, USA; niraj.vora@bswhealth.org; 6Department of Medical Physiology, Texas A&M University School of Medicine, Bryan, TX 77807, USA

**Keywords:** pre-eclampsia, marinobufagenin, cytotrophoblast, placenta, anti-angiogenic, monoclonal antibody

## Abstract

**Background/Objectives**: Marinobufagenin (MBG) is a biomarker that is found to be high in pre-eclampsia (preE), and thus is relevant in the pathogenesis of obstetric complications. MBG is thought to possibly be implicated in harmful signaling within cytotrophoblasts (CTBs) of the placenta. In this study, we evaluated how anti-MBG human monoclonal antibody can alter cellular signaling in CTBs and in a rat model of preE. **Methods**: CTB cell proliferation, migration, and invasion as a result of MBG, both with and without anti-MBG present, were monitored via cell-based studies. Pro-angiogenic and anti-angiogenic factors in response to MBG with and without antibody were measured. Finally, we evaluated the lead anti-MBG antibody in comparison with the parent murine antibody in a rat model of preE. **Results**: CTB cells exposed to ≥1 nM MBG showed decreased (*p* < 0.05) proliferation, migration, and invasion, decreased secretion of VEGF and PIGF, and increased secretion of sFlt-1 and sEng. Pretreatment with anti-MBG significantly (*p* < 0.05) attenuated MBG-induced CTB dysfunction and modulation of VEGF, PIGF, sFlt-1, and sEng expression. In the rat model, anti-MBG treatment normalized blood pressure, reduced proteinuria, and eliminated fetal effects. **Conclusions**: MBG is a potential causative agent for preE, as it causes dysfunction in CTBs due to anti-angiogenic milieu. Our study suggests that anti-MBG antibody binds to MBG, neutralizing it and preventing downstream signaling in vitro. In a rat model of preE, treatment with anti-MBG antibody was effective at normalizing blood pressure, kidney function, and fetal birth weights. These data suggest that a human monoclonal antibody with high specificity and affinity for MBG has potential as a therapeutic agent for preE.

## 1. Introduction

Pre-eclampsia (preE) is the second most frequent cause of maternal and fetal morbidity and mortality, accounting for about 60,000 maternal deaths annually worldwide [1]. PreE is characterized by hypertension (diastolic ≥ 90 mm Hg) and substantial proteinuria (≥300 mg in 24 h) after 20 weeks of gestation [2]. It occurs in 5–10% of pregnancies [3], and its incidence is on the rise in the US [4,5]. The pathophysiological triggers and mechanisms of preE are not well characterized [6]. There is no reliable biomarker for its early diagnosis, and no definitive therapy other than delivery. Therefore, this condition represents an unmet medical need.

Vasoconstrictors such as digitalis-like cardiotonic steroids (CTSs) are a second natriuretic system. In addition to vaso-relaxant atrial natriuretic peptides [7], CTSs are key factors in blood pressure regulation. Several CTSs have been identified in human plasma and urine, including cardenolides (e.g., endogenous ouabain [EO]) and bufadienolides (e.g., marinobufagenin [MBG]). Binding to the CTS receptor site on the α-subunit of Na^+^/K^+^ ATPase (NKA) induces natriuresis [8,9,10]. MBG, but not EO, is a potent vasoconstrictor [11], impairing nitroprusside-induced relaxation of umbilical arteries [12]. In addition to the transport of Na^+^ and K^+^, NKA functions as a receptor, which is capable of transducing CTS binding into activation of intracellular protein kinases and alterations in Ca^2+^ levels, ultimately altering the cell surface expression of the NKA and Na^+^/H^+^ exchanger [13]. MBG participates in EGFR-dependent cell signaling, which induces oxidative stress and promotes fibrosis [14]. In light of this, CTSs can be viewed as a new class of steroid hormones, rather than simply as endogenous inhibitors of Na^+^ transport. Recent data support a role for CTS, specifically MBG, in the pathogenesis of hypertension and preE. Normal plasma MBG (0.225 nM) is drastically elevated in essential hypertension and congestive heart failure, as well as in chronic renal failure [9,10].

MBG, but not endogenous ouabain, is markedly elevated in preE [15,16,17]. Puschett and co-workers developed an ELISA with high specificity for MBG [15], which revealed a 5-fold increase in serum MBG and a 4-fold increase in urine MBG levels in preE patients vs. normotensive pregnant women. Plasma levels of MBG, but not EO, become elevated in patients with preE [15,16,17]. We demonstrated that the MAPK signaling is involved in the deleterious effects of MBG on cytotrophoblast (CTB) function, which is important for normal placental development [18,19,20]. High blood levels of MBG during pregnancy may directly contribute to preE pathogenesis, in part through detrimental cellular signaling in CTB cells [21,22]. A syndrome with many phenotypic characteristics of preE results when pregnant rats are given weekly injections of desoxycorticosterone acetate (DOCA), and their tap water is replaced with normal saline [23]. This syndrome can be reproduced by daily injections of MBG, beginning in early pregnancy [23,24]. Angiogenic imbalance played a role in the pathogenesis of preE in this rat model; however, a relevant earlier event appears to be the elaboration and secretion of MBG [25]. If MBG and other CTSs promote preE pathology, reversing the effects of MBG by scavenging or competition represents a reasonable treatment approach. Digibind and DigiFab are polyclonal anti-digoxin antibodies with some cross-reactivity for other CTSs. Both of these polyclonals have been shown to reduce circulating levels of CTSs. In animal models, Digibind reverses MBG-induced vasoconstriction by restoring the activity of erythrocyte Na/K-ATPase, a target enzyme for CTS [26,27]. Recently, it was shown that plasma levels of endogenous MBG are related to salt sensitivity in men [28]. Also, magnesium can increase the efficacy of immunoneutralization of MBG-induced NKA inhibition in erythrocytes ex vivo [29]. It was shown that DigiFab interacts with CTSs from preE plasma and reverses preE-induced NKA inhibition. Despite their very limited reactivity with MBG, both Digibind and DigiFab have been explored for the treatment of patients with preE [30]. Digibind treatment lowered blood pressure and reduced proteinuria in a preE rat model [25]. It has been reported to improve or slow progression of preE in a few small clinical trials [26,27,31]. RBG is a structural congener of MBG that acts as a competitive antagonist of MBG. We observed that when RBG was administered early in pregnancy, it prevented the manifestations of preE in a rat model of preE [32,33,34]. Conversely, when RBG was given to normally pregnant rats, they developed preE syndrome [32]. In a preliminary study, Vu et al. administered a polyclonal MBG antiserum (MBG-P, Ab) in a rat model of preE (PDS) on the 16th, 17th, and 18th day of pregnancy, and demonstrated that the blood pressures of PDS rats returned to normal [24]. Fedorova et al. reported similar data, where a mouse monoclonal Ab to MBG (3E9) decreased blood pressure in expectant rats that were artificially made hypertensive by feeding them high-salt diets [31]. Due to the potential for immunogenicity of polyclonal and murine antibodies, human monoclonals are much preferred for use in pregnant patients.

We have identified a novel anti-MBG human monoclonal antibody for potential use as an innovative, effective, and safe therapeutic for preE. As the focus of this work is on developing a blockade of MBG effects in pregnancy, we plan to assay MBG in patients at high risk for preE to test the hypothesis that MBG increases prior to preE development. We and others have already shown that MBG is elevated once preE has been diagnosed [15,16,17].

In addition, we have conducted a preliminary study of the efficacy of the antibody in a desoxycorticosterone (DOCA)-saline rat model of pre-eclampsia. The DOCA model is a volume-expanded model of preE in pregnant rats [23,24], where saline is switched in for normal drinking water, and a deoxycorticosterone acetate (DOCA) injection induces volume expansion, thus raising plasma and urine MBG levels in a way that resembles characteristics of human preE, including proteinuria, hypertension, and intrauterine growth restriction (IUGR), where pup number and litter weight are low [23,24,32].

These data will allow us to assess the potential of anti-MBG monoclonal antibody therapy to intervene in the pathophysiological cascade of preE.

## 2. Materials and Methods

Anti-MBG human monoclonal antibodies: A human phagemid library with 1.2 × 10E10 independent clones were panned against MBG that was further conjugated to bovine serum albumin (BSA). Several strategies were employed to enrich for MBG binding clones as the panning proceeded through 4 rounds of enrichment, including linking multiple CTS competitive elution profiles together to increase the likelihood for high-affinity MBG-only binders. Approximately 660 individual clones, representing 6 different pan-enriched fractions, were tested for binding to MBG in ELISA, and ~110 putative binders were further analyzed by BstN1 fingerprinting, DNA sequencing, and/or MBG competitive ELISA. These were reduced to ~14 unique clones, of which 7 were converted to human IgG1 for testing in cell-based MBG neutralization assays. Clones representative of distinct CTS competitive elution profiles were also characterized for their binding profile on MBG-BSA in the presence of various CTSs (MBG, RBG, cinobufagin-CINO, ouabain-OUB, digoxin-DIG), either as phage, soluble Fab, or IgG. Initial studies were conducted with these phage-derived antibodies.

A second strategy was to humanize the previously described 3E9 murine mAb [31]. Humanization was performed using the modification method of Queen et al. [35]. The complementarity determining regions (CDRs) of the light (kappa) and heavy (IgG1) chains of the mouse antibody were grafted and transferred onto acceptor human sequence frameworks, with the framework itself passing as the segment of the variable regions that encompassed everything but the CDRs. Human acceptor frameworks were built by aligning the mouse framework sequences against a database of human framework sequences (see below), in an attempt to find the closest human homolog for each chain (that usually has 65–70% sequence identity). Three potential heavy chains (H1, H2, and H3) as well as two potential light chains (L1 and L2), were made. The differences among these variants reflect the possible importance of retaining some mouse framework amino acids that may be involved in binding. The proposed designs show equivalent-to-better similarity to human germlines (87–95%) than the set of FDA-approved human antibodies. A mix-and-match strategy yielded six combinations of the humanized heavy and light chains, in an attempt to determine the best variant. We cultured 293F cells (Invitrogen) in serum-free medium (Freestyle, Invitrogen) in 6-well plates, and co-transfected them with various combinations of heavy- and light-chain plasmids at a 1:1 DNA ratio. Transfections were carried out using polyethyleneimine. On day 3 post-transfection, cell culture supernatants were harvested. Antibody concentrations in the supernatants were estimated from SDS-PAGE gels. Based on developability criteria [20], the humanized anti-MBG monoclonal antibody H3L2 was ultimately chosen as the lead candidate for further preclinical studies.

Cell proliferation assay: Vascular endothelial damage is part of the pathology seen in preE, and MBG promotes endothelial dysfunction in part by inhibition of proliferation, as described previously by Uddin et al. [20]. Human umbilical cord venous endothelial cells (HUVECs, ATCCs) in endothelial cell growth medium were plated onto 96-well plates. The next day, the cells were pretreated with various antibodies (2.5–20 μg/mL), and then incubated with 2 nM MBG for 48 h. ^3^H-thymidine was added to each well. ^3^H-thymidine-labeled DNA was assayed after harvesting the cells using a PerkinElmer Wallac Trilux liquid scintillation counter. The mouse anti-MBG mAb 3E9 served as a positive control, and a non-MBG binding human IgG mAb served as a negative and isotype control. Means were tested for significance using Student’s *t*-test.

Human CTB proliferation assay: Human CTBs were treated with DMSO (vehicle) or 0.1, 1, 10, or 100 nM of MBG for 48 h. Some cells were pretreated with anti-MBG monoclonal antibodies H3L2 for 2 h. Culture media were collected for analysis of pro-angiogenic and anti-angiogenic factors. Cell viability was measured using a CellTiter Assay (Promega). Levels of vascular endothelial growth factor (VEGF), placental growth factor (PlGF), sFlt-1, soluble endoglin (sEng), and IL-6 were measured in culture media by ELISA. Statistical comparisons were performed using analysis of variance with Duncan’s post hoc test.

Human CTB Migration Assay: Human CTBs were treated with DMSO (vehicle) or 0.1, 1, 10, or 100 nM of MBG for 48 h. Some cells were pretreated with anti-MBG monoclonal antibodies H3L2 for 2 h. Cell migration was measured using a CytoSelect Assay (Cell Biolabs, San Diego, CA, USA), as described previously by Uddin et al. [20]. The CytoSelect Cell Migration Assay Kit includes polycarbonate membrane inserts (8 μm pores) in a 24-well plate, where the membrane enables migratory cells to sift through and advance towards a chemoattractant (EGF, 10 ng/mL) in the lower well. Cells (0.5–1.0 × 10^6^ cells/mL) were treated with DMSO (control) or MBG (10 and 100 nM), and incubated for 24 h, after which migratory cells were detached using Cell Detachment Solution. Afterwards, they were lysed and stained with the CyQuant GR Dye solution, which was added to the cells. Fluorescence was then measured at 480 nm/520 nm using a plate reader (CytoFluor Series 4000 Fluorescence Multi-Well Plate Reader, (Applied Biosystem, Waltham, MA, USA) to quantify the amount of cell migration.

Human CTB Invasion Assay: Human CTBs were treated with DMSO (vehicle) or 0.1, 1, 10, or 100 nM of MBG for 48 h. Some cells were pretreated with anti-MBG monoclonal antibodies H3L2 for 2 h. Cell invasion was measured using EGF-induced invasion, determined using the quantitative FluoroBlok invasion assay, as described previously by Uddin et al. [20]. The FluoroBlok invasion assay uses Matrigel to mimic the extracellular matrix. Serum-starved SGHPL-4 cells were pretreated with DMSO (control) or MBG (10 and 100 nM) for 2 h, trypsinized, and then planted onto Matrigel-coated 8-μm FluoroBlok membrane inserts (2.5 × 10^5^ cells/insert) that were then placed in a 24-well plate with EGF (10 ng/mL) and incubated for 20 h. This enabled invasion. Invaded cells were fluorescently labeled with Calcein-AM and detected using a fluorescence plate reader which blocked light (490–700 nm), thus ensuring only invaded cells were measured.

Measurement of angiogenic factors: Levels of the pro-angiogenic factors VEGF and PlGF, and the anti-angiogenic factors sFlt-1 and sEng, were measured with commercially available kits (Human VEGF Quantikine ELISA Kit (DVE00); Human PlGF Quantikine ELISA Kit (DPG00); Human sVEGF R1/Flt-1 Quantikine ELISA Kit (DVR100B); Human Endoglin/CD105 Quantikine ELISA Kit (DNDG00); R&D Systems, Minneapolis, MN, USA).

DOCA-saline rat model of pre-eclampsia: Female Sprague Dawley rats (200–250 g, Charles River) were mated with male Sprague Dawley rats (275–300 g). Pregnancy was confirmed via vaginal smears or by using vaginal plugs.

The rats were separated and studied in 5 groups (*n* = 10 per group): Group 1 (NP): normal pregnant rats; Group 2 (PDS): gravid rats initially injected i.p. with 12.5 mg of DOCA in a depot form, followed by a weekly i.p. injection of 6.5 mg of DOCA, as well as with drinking water replaced with 0.9% saline; Group 3 (NPM): normal gravid rats administered MBG injections everyday (7.65 µg/kg/day) after pregnancy was established on day four; Group 4 (PDS-3E9): rats given DOCA and saline, similarly to Group 2, as well as given the murine anti-MBG antibody 3E9 (2.2 mg/kg/day) from GD16-GD18; Group 5 (PDS-H3L2): rats given DOCA and saline, similarly to Group 2, as well as humanized anti-MBG antibody H3L2 (2.2 mg/kg/day) from GD16-GD18. The doses of antibody concentrations (2.2 mg/kg = 3.7 nmol for a 0.25 kg rat) were determined to be sufficient to neutralize anticipated levels of plasma MBG [15].

Systolic blood pressure (BP) was measured with a tail cuff on days 17, 18, and 19. 3 BP readings were taken and the mean was then calculated, so the mean was used as the BP value. On days 18–20 of pregnancy, 24 h urine was collected without food, to avoid excess protein contamination in urine from any fallen food particles. Each rat was harbored separately in a metabolic cage. The 24 h protein excretion was measured and was normalized to creatinine. Blood samples of the rats were collected after the last measurement on GD18-20. The number of fetal pups was counted to find the mean number of pups, along with examining any developmental abnormalities of the pups. Statistical analysis was performed using analysis of variance and Tukey’s post hoc test, with a *p* value ≤ 0.05 being considered statistically significant.

The animal studies were conducted with the approval of the IACUC of Texas A&M Health Sciences University/Baylor Scott & White (IACUC number: 110500; IACUC approval date: 18 August 2017), in conformance to the National Institutes of Health guide for the care and use of laboratory animals.

## 3. Results

Anti-MBG antibodies: The relative specificity for MBG of our lead phage antibody, 201/202, and the two back-up mAbs, 206/208 and 236/237, is shown in Figure 1. Binding of 201/202 to immobilized MBG was best inhibited by free MBG, with partial inhibition by RBG and CINO. In contrast, DIG and OUB did not compete for binding at all. Table 1 summarizes the binding avidity and specificity of the lead candidate and the two alternatives. MBG was conjugated to BSA or KLH to facilitate coating of plates. All three mAbs showed low binding to BSA or KLH and minimal binding to RBG and related CTS.

HUVEC proliferation: Anti-MBG humAbs prevented MBG inhibition of cultured human umbilical cord venous endothelial cell (HUVEC) proliferation (Figure 2).

Humanized anti-MBG antibody H3L2 attenuated MBG-induced dysfunction of CTB cells: CTB cells exposed to MBG at levels of 1 nM or greater had decreased (*p* < 0.05) proliferation (Figure 3A), migration (Figure 3B), and invasion (Figure 3C). Pretreatment with H3L2 significantly (*p* < 0.05) attenuated the MBG-induced downregulation of CTB cell proliferation (Figure 3A), migration (Figure 3B), and invasion (Figure 3C).

Humanized anti-MBG antibody H3L2 attenuated MBG-induced anti-angiogenic milieu in CTB cells: MBG at levels of 1 nM or greater decreased secretion of VEGF and PIGF (*p* < 0.05; Figure 4A,B) and increased secretion of sFlt-1 and sENG (*p* < 0.05; Figure 4C,D) by CTB cells. Pretreatment with H3L2 significantly (*p* < 0.05) attenuated the MBG-induced modulation of angiogenic VEGF and PIGF (*p* < 0.05; Figure 4A,B) and anti-angiogenic factors (*p* < 0.05; Figure 4A,B).

Efficacy of anti-MBG antibodies in the DOCA-saline rat model: DOCA and saline treatments resulted in characteristics of preE, like increased BP, proteinuria, and an overall reduction in litter size, as well as fetal malformations in pregnant rats. MBG alone also had similar effects [24]. Anti-MBG murine antibody 3E9, however, prevented these increases in blood pressure, proteinuria, and fetal malformations. Humanized anti-MBG antibody H3L2 also produced a similar effect to the parent murine antibody 3E9 (Table 2).

Baseline (initial) blood pressure and urine protein did not significantly differ among treatment groups. Animals treated with DOCA and saline (NPS) showed statistically significant spikes in BP, proteinuria, and percentage of malformed pups, in comparison to control pregnant animals (NP) (*p* < 0.05). Animals treated with MBG alone (no DOCA or saline) (NPM) also showed statistically significant spikes in BP, proteinuria, and percentage of malformed pups, relative to control pregnant animals (NP) (*p* < 0.05). Animals treated with anti-MBG murine antibody 3E9 or with anti-MBG human antibody H3L2 did not show a statistically significant increase in BP, proteinuria, or malformed pups, compared to control DOCA-saline-treated animals (PDS). Multiple controls were included in the study. Normal pregnant rats served as a negative control for the effect of MBG or saline treatment. MBG-treated rats and DOCA/saline-treated rats served as a positive control for the development of preE symptoms. Animals treated with the murine antibody 3E9 served as a positive control for the reduction in preE symptoms in the model. Based on our own previous experience and prior publications [24], 3E9 is expected to reduce blood pressure, proteinuria, and fetal malformations.

## 4. Discussion

The results of our study add to several decades of research implicating MBG as a causative agent for preE. Here, we demonstrate dysfunction in CTB cells due to an anti-angiogenic milieu produced by MBG treatment. Proper placental formation requires adequate CTB cell invasion in the endometrium. In preE, proper differentiation of CTB cells is obstructed, leading to abnormal placental formation, culminating in an array of issues for the pregnancy and the fetus [20]. Due to preE, the extent of CTB cell invasion in the uterine lining is greatly affected, and consequently, a reduction in uteroplacental perfusion results in placental focal ischemia and hypoxia [20]. Elevated plasma levels of MBG are seen in patients with preE [15,16,17]. Uddin et al. saw that MBG increased microvascular barrier permeability in an animal model of preE [18], and preE patients have been shown to demonstrate increased vascular permeability [18,36,37]. In addition, they determined that MBG-induced impairment of CTB cell function was a result of decreased ERK1/2 activity, and MBG treatment drastically affected the growth factor-induced migration and invasion of CTB cells [20]. MBG interferes with CTB function and may cause defective placentation, resulting in a lack of vascular remodeling. Consequently, hypoperfusion of the maternal–fetal unit is possible. MBG causes hypoxia and ischemia, leading to continued elevation of MBG levels and an angiogenic imbalance. CTB dysfunction is mediated by alterations in signaling pathways that stimulate apoptotic and stress signaling. These changes culminate in the production of endothelial dysfunction and oxidative stress, leading to the induction of preE syndrome [18].

Fedorova et al. hypothesized MBG to be a potential target for immunoneutralization in preE [31]. Immunoassay based on 4G4 anti-MBG mAb revealed a three-fold increase in renal excretion of MBG in hypertensive Dahl-S rats. Their experimental findings revealed that 3E9 anti-MBG mAb decreased blood pressure and restored vascular sodium pump activity [31]. Uddin et al. recognized RBG as a potential compound to negate the effects of MBG-induced preE, since RBG acts as a competitive antagonist congener of MBG [32,38]. It is known to prevent the development of hypertension, proteinuria, and intrauterine growth restriction (IUGR) in early pregnancy. MBG is known to downregulate aortic AT1 receptor expression, leading to multiple downstream reactions, ultimately causing CTB dysfunction. Western blot results showed that RBG treatment increased expression of AT1 receptor, thus attenuating the negative effects of MBG [32]. Digibind antibody, a polyclonal, anti-digoxin Fab with some cross-reactivity to MBG, could reverse preE syndrome in a rat model, alleviate inhibition of NKA by MBG, and simultaneously normalize blood pressure in preE patients. Immunoneutralization of MBG by Digibind or treatment with RBG can attenuate the development of preE in animal models, providing hope that treatments aimed at interfering with MBG-induced pathophysiology can be effective.

Excessive volume expansion during the first trimester of pregnancy elevates circulating levels of MBG, which causes an increase in the expression of RAS, including AT1 receptor. RAS is known to be naturally present in the endometrial region, and is known to function in uteroplacental blood flow and vascular remodeling during pregnancy. Due to upregulation of RAS, oxidative stress increases, leading to endothelial dysfunction. MBG also induces fibrosis and contributes to vascular stiffening in preE. This pro-fibrotic effect has been demonstrated in renal and cardiovascular tissues, and more recently, in preE [12,39,40]. Recent data strongly support the involvement of MBG in preE [14,17]. CTSs, “endogenous digoxin-like factors” (EDLF), have been known since the 1980s to increase significantly during pregnancy-induced hypertension and preE [27,41,42].

In this study, we demonstrated that anti-MBG antibodies can reverse the anti-angiogenic milieu in CTB cells and restore the invasive capacity of CTB cells for proper placental formation (Figure 5). Figure 5 illustrates that in Phase I, early in the first trimester, excessive volume expansion during pregnancy can increase MBG levels. This has several pernicious downstream effects—with one of the major ones being damage to cytotrophoblast functions later in the pregnancy. Because functioning cytotrophoblasts are needed for a healthy placenta—which is the vessel for blood supply to fetuses—increased MBG levels can consequently upset the angiogenic balance and increase free radicals and oxidative stress that cause inflammation, eventually leading to pre-eclampsia in the second trimester, as shown in Phase III. Pre-eclampsia leads to many negative maternal and fetal outcomes, as it can cause high BP, proteinuria, and intrauterine growth restriction, among several other adverse outcomes. As illustrated, early on in the pregnancy, problems start to crop up in early–mid-first trimester in Phase 1. Thus, if we can find a way to scavenge the MBG that increases in early–mid-first trimester—whether with antibodies or other binding therapies—we can stymie the progression of all the deleterious downstream effects that are induced as a result in the later trimester. By neutralizing MBG early on in pregnancy, pre-eclampsia in the second trimester can be revolutionarily halted in its tracks, drastically improving maternal and fetal health outcomes. We further showed that anti-MBG human monoclonal antibody H3L2 can reverse hypertension, proteinuria, and fetal effects in a rat model of preE, even when given late in pregnancy. We note that the time of administration in the DOCA-saline model corresponds to the second trimester of human gestation, illustrating a shortcoming of rodent models of preE. Non-human primate models are needed to facilitate translational studies of this and other therapies.

## 5. Conclusions/Perspectives

A digitalis-like factor, marinobufagenin (MBG), has been implicated as a causative factor in pre-eclampsia (preE). A human monoclonal antibody (humAb) that binds to MBG with high affinity and specificity has been identified. The anti-MBG antibody has great potential for therapeutic use. The data of this study suggest that anti-MBG antibody binds to MBG, neutralizing it and preventing downstream signaling in vitro. In a rat model of preE, treatment with anti-MBG antibody was effective at normalizing blood pressure, kidney function, and fetal birth weights. These data provide a strong foundation for further preclinical development of this antibody and its eventual testing in the clinic. By analogy to other approved therapeutic antibodies, whose half-lives range up to 4 weeks [37], this antibody will likely have a plasma half-life in humans of approximately 3 weeks. Thus, we envision a single course of treatment, initially for high-risk patients, to enable continuation of pregnancy toward a normal delivery time, with improved maternal and neonatal outcomes.

## Figures and Tables

**Figure 1 biomedicines-13-00782-f001:**
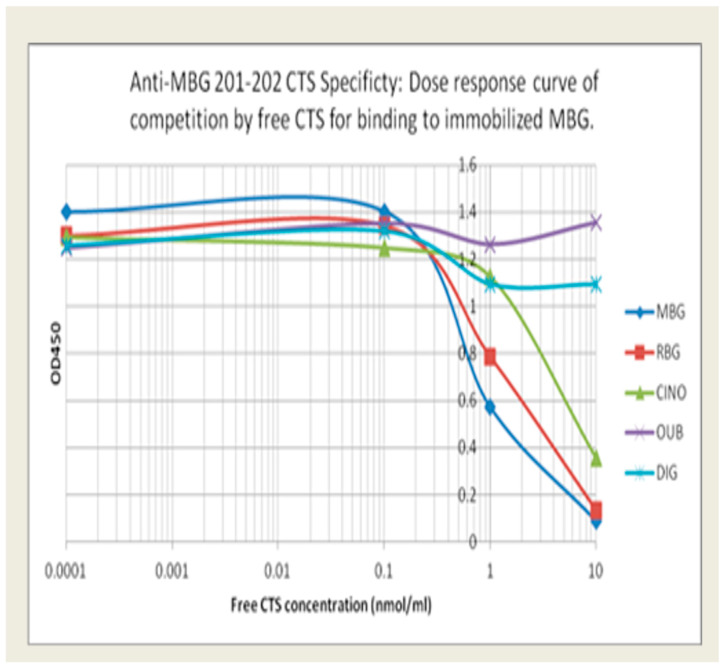
The relative binding specificity of 201/202 to immobilized MBG was best inhibited by free MBG, with partial inhibition by RBG and CINO. In contrast, DIG and OUB did not compete for binding.

**Figure 2 biomedicines-13-00782-f002:**
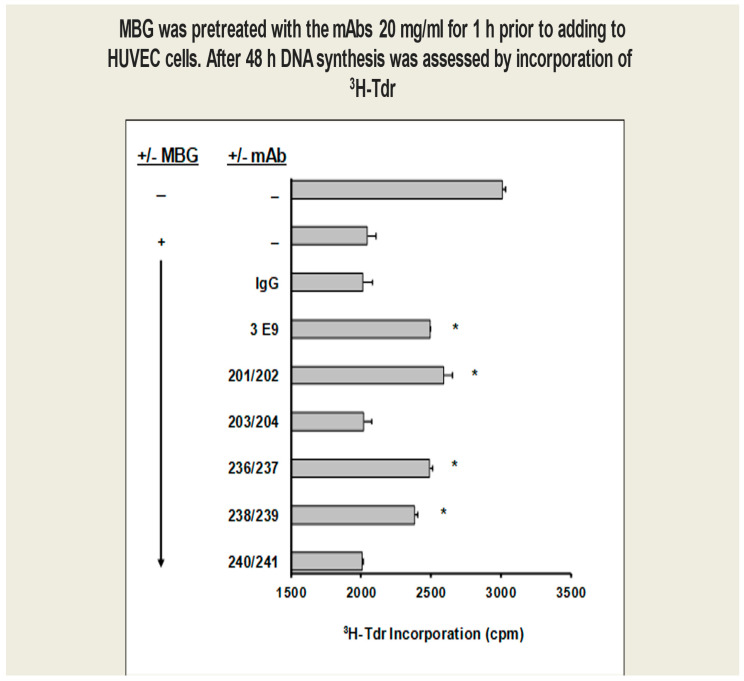
MBG-induced proliferation of cultured human umbilical cord venous endothelial cells (HUVECs) was attenuated by anti-MBG mAbs (* *p* < 0.05). MBG inhibition of HUVEC proliferation was significantly reduced by 201/202, 236/237, and 238/239 among the phage-derived human mAbs, with 201/202 being the most active.

**Figure 3 biomedicines-13-00782-f003:**
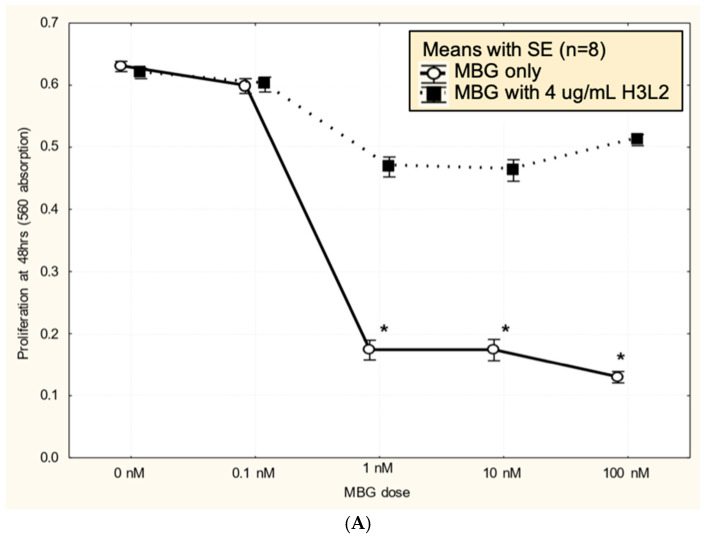
(**A**). MBG at a concentration ≥ 1 nM decreased cytotrophoblast cell proliferation (* *p* < 0.05), and pretreatment with H3L2 significantly attenuated the MBG-induced downregulation of cell proliferation (* *p* < 0.05). (**B**). MBG at a concentration ≥ 1 nM downregulated cytotrophoblast cell migration (* *p* < 0.05), and pretreatment with H3L2 significantly attenuated the MBG-induced downregulation of CTB cell migration (* *p* < 0.05). (**C**). MBG at a concentration ≥ 1 nM decreased cytotrophoblast cell invasion (* *p* < 0.05), which was significantly attenuated by pretreatment with H3L2 (* *p* < 0.05).

**Figure 4 biomedicines-13-00782-f004:**
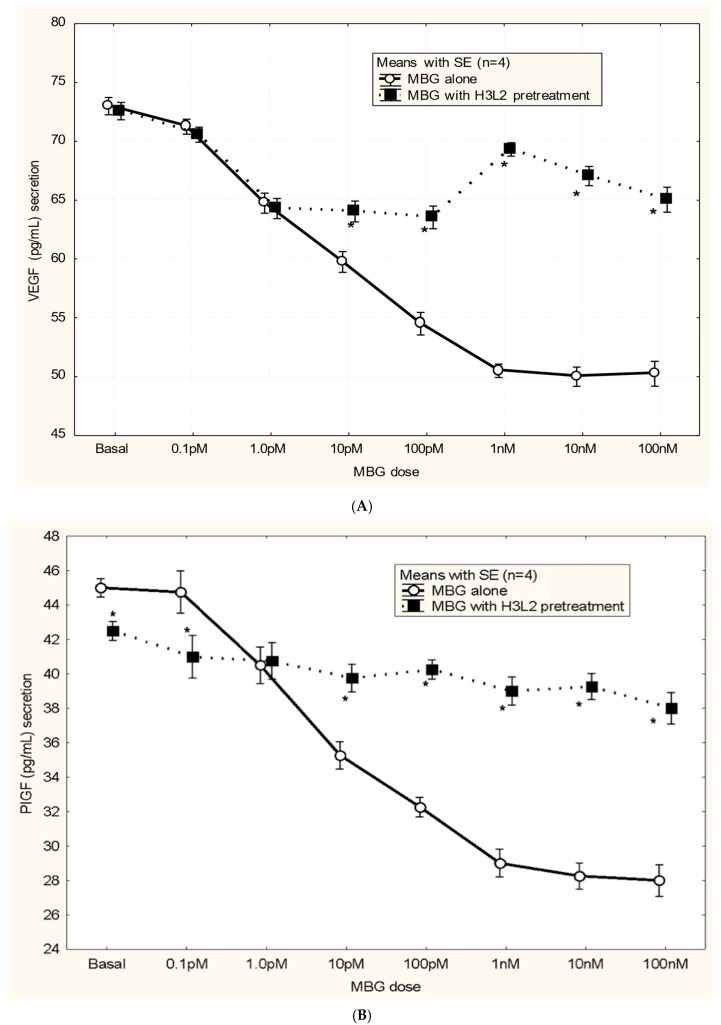
(**A**) MBG-induced decreased expression of VEGF was attenuated by pretreatment with H3L2 (4 µg/mL) (* *p* < 0.05). (**B**) MBG-induced decreased expression of PlGF was attenuated by pretreatment with H3L2 (4 µg/mL) (* *p* < 0.05). (**C**) MBG-induced increased expression of sFlt-1 was attenuated by pretreatment with H3L2 (4 µg/mL) (* *p* < 0.05). (**D**) MBG-induced increased expression of sEng was attenuated by pretreatment with H3L2 (4 µg/mL) (* *p* < 0.05).

**Figure 5 biomedicines-13-00782-f005:**
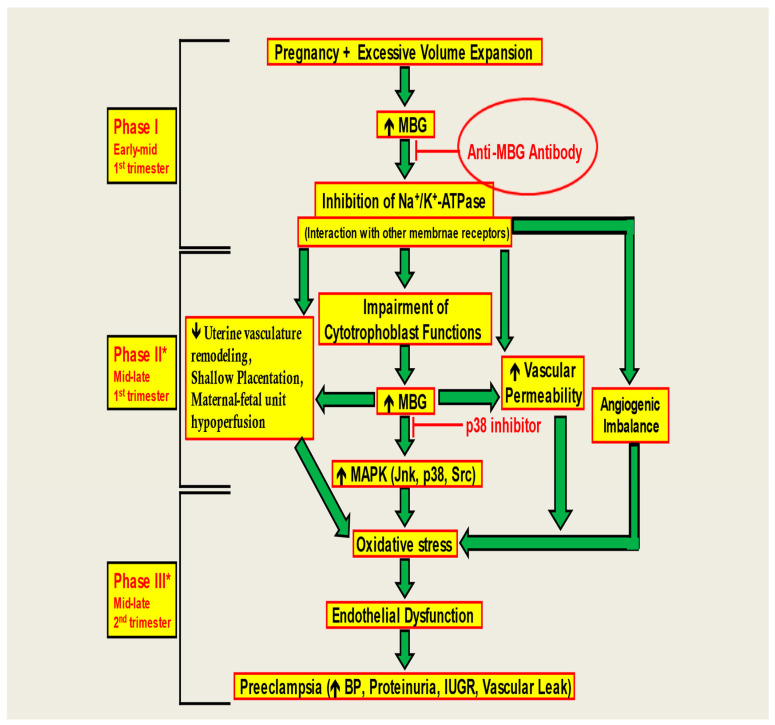
A model of the negating effect of anti-MBG antibodies to attenuate the anti-angiogenic milieu and prevent preE in CTB cells. The monoclonal anti-MBG antibody binds to the MBG, causing an immunoneutralization reaction, preventing the downstream reactions in vitro.

**Table 1 biomedicines-13-00782-t001:** Estimated avidity (KA) of human anti-MBG antibodies (antibody concentration at 50% maximal binding to antigen). Relative specificity of anti-MBG antibodies for CTS in competition with MBG (immobilized) at 50% maximal binding. ^1^ Average of 3 independent titrations. ^2^ Average of 2 independent titrations. ^3^ Average of 4 independent titrations.

Antibody	Antigen 1 (MBG-BSA)	Antigen 2 (MBG-KLH)	Relative CTS Specificity as % of MBG Binding (MBG-RBG-CINO-OUB-DIG)
201–202	0.33 nM ^1^	0.20 nM ^2^	100-50-20-0-0
206–208	1.35 nM ^3^	0.20 nM ^2^	100-20-0-0-0
236–237	0.12 nM ^2^	0.14 nM ^2^	100-0-0-0-0

**Table 2 biomedicines-13-00782-t002:** Anti-MBG effects in the MBG-induced pre-eclampsia rat model. Administration of either exogenous MBG or saline drinking water yielded significant spikes in blood pressure (BP) and urinary protein, along with a reduction in litter size, but an increase in malformed pups. Anti-MBG antibody rescued the normal phenotype in pregnant rats on saline water.

Groups	Blood Pressure—Initial (mm Hg) *	Blood Pressure—Final (mm Hg) *	Urine Protein (mg/24 h) **	No. of Pups	% Malformed Pups
NP	103 ± 5	98 ± 7	2.8 ± 1.1	14 ± 1.8	0
PDS	101 ± 8	141 ± 9 *	5.7 ± 1.6 *	10.2 ± 1.5 *	16 *
NPM	104 ± 4	137 ± 5 *	6.4 ± 1.8 *	9.5 ± 1.3 *	18 *
PDS-3E9	102 ± 8	100.8 ± 8	3.1 ± 0.9	13.6 ± 2.1	0
PDS-H3L2	104 ± 5	109.4 ± 11	3.5 ± 0.7	12.9 ± 3.2	0

NP: normal pregnant. PDS: NP + saline drinking water (DOCA model of preE); N = 10 for all treatment groups. NPM: NP + MBG (normal pregnant rats given daily injections of MBG (7.65 µg/kg/d) once pregnancy was established on day four of the experiment). PDS-3E9: PDS + anti-MBG murine antibody (PDS rats injected i.p. with anti-MBG antibody 3E9, 2.2 mg/kg/d; on GD16, GD17, and GD18). PDS-H3L2: PDS + anti-MBG humanized antibody (PDS rats injected i.p. with anti-MBG antibody H3L2, 2.2 mg/kg/d; on GD16, GD17, and GD18). * Blood pressure measured by tail cuff. ** Urine protein normalized to creatinine.

## Data Availability

The original contributions presented in this study are included in the article. Further inquiries can be directed to the corresponding author.

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
