# Peer review of "Neutralization of Marinobufagenin Demonstrates Efficacy In Vitro and In Vivo in Models of Pre-Eclampsia"

_biomedicines, 2025, doi:10.3390/biomedicines13040782_

Round 1
Reviewer 1 Report
Comments and Suggestions for Authors
The study presents a very important topic of potential therapeutic antibodies for the treatment of preeclampsia. The paper is of high interest however there are some major issues that need to be fixed prior to publication:
Minor and major issues:
In the Abstract , rephrase part of the sentence “eliminated fetal effects” or provide more info which effects. Instead of “potential causative agent” rather put “among key factors in the pathogenesis of preE”
In line 86, add sentence “In a rat model of the syndrome, MBG induces hypertension, proteinuria, intrauterine growth restriction, and increased weight gain, but these symptoms are prevented by an MBG antagonist.( · DOI: 10.1016/j.bbadis.2010.02.005 ),”
Lines 170-171 should be transferred elsewhere e.g. in Disscussion: “Vascular endothelial damage is part of the pathology seen in preE, and MBG promotes endothelial dysfunction in part by inhibition of proliferation as described previously by Uddin et al [22, 23].” Instead, provide here the procedure for cultivation of HUVECs, and provide reference for 3H-thymidine proliferation assay. Also, add the reference that describes that pre-treatment protocol for 1h anti-MBG humAbs is sufficient. Provide info on how you selected MBG (2 nM) concentration for this assay.
Next, describe the cultivation of CTB cells.Are Human CTBs primary cells used or cell line?
In line 179 the Authors mention they collected culture media for analysis of pro-angiogenic and anti-angiogenic factors. Add term “secreted protein” pro-angiogenic or angiogenic factors.
Line 185 “Some cells were pretreated”, describe in more details which cells.
Provide more details for migration and invasion assays. Were the cells serum-starved, at what time point the scratch was made and how long did you incubate the cells after scratching, did you use chemoattractant ?
Line 211 add details: How did you choose concentrations for the treatment of rats with H3L2 (2.2 mg/kg/day)and anti-MBG antibody 3E9 (2.2 mg/kg/day)?
Figure 2 should be corrected. In the methodology you stated 20 μg/mL concentration of MBG and in chart you wrote 20 mg/mL. Avoid the too descriptive title of Fig 2, instead the title should be “Proliferation of MBG-induced HUVEC cells and the effect of anti-MBG mAbs”. Figure legend should also be corrected, not to narratively describe the result but to provide information about details on the chart and used methodology, number of replicate experiments, compared to what you expressed the p values etc. Also, the explanation of the results in the section 258-259 is poor and lacks the description of the effects of specific tested anti-MBG mAbs.
Line 263 Add at the end of sentence compared to what you expressed the significant difference.
Again, the description of the results of anti-MBG antibody H3L2 effects on CTB migration, proliferation and invasion are very poor, with insufficient information.
Figure 4C is of insufficient quality.
Table 2 title is too descriptive, please correct properly. Text from 320-324 should be placed not in the title, but in the description of the results.
Also, in table 2 add in the legend what the* stands for, the difference compared to what group? Is every other group being compared to PDS group? Description in the lines 338-341 should be in the table legend explaining these comparisons.
There is no representation of baseline urine protein concentration in the Table2, and yet you describe the result in text in line 328. Please add values in the Table 2.
Line 347 rephrase “as a causative agent” since it is not a single causative factor for preE, but rather state “among key factors”
Line 356 should be a separate sentence since it refers to human studies, while line 355 refers to animal models of preE.
Lines 359 to 362 should be removed since they describe the phenomena already mentioned at the beginning of the same paragraph.
Line 373” It is known to prevent the development of hypertension, proteinuria, and intrauterine growth restriction (IUGR) in early pregnancy” Provide reference and explanation is this animal study, are you referring to reference 49?
In line 380 at the end of sentence you state “normalizes blood pressure in preE patients” and in the same sentence you mention rat model. Is this a human study or animal study? Provide reference for this statement.
Lines 384-389 provide references.
In lines 399 to 400 the authors state : “We note that the time of administration in the DOCA-saline model corresponds to second trimester of human gestation, illustrating a shortcoming of rodent models of preE.” Provide more discussion on this mechanisms that you represented in the illustration no 5.
Also, at the end of the Discussion provide more description on the limitations of the study.
In Conclusion you state that anti-MBG antibody binds to MBG, neutralizing it,and preventing downstream signaling. But in this work, you did not analyze signaling pathways, but functional responses of CTB cells and secretory angiogenic response of HUVEC cells. Rephrase the first sentence of Conclusions and mention these actually observed results.
Lines 413-417 should be removed from the conclusion, since this was not evaluated in this paper. You can state this at the end of Discussion as a sort of future prospects.
Author Response
Summary of revisions to manuscript: biomedicines-3480491
We would like to express our appreciation to both reviewers for their insightful comments and suggestions, which improve both the premise, conclusions, and presentation of our study. A summary of our revisions follows, organized by our response to each critique.
Reviewer 1
Critiques: In the Abstract, rephrase part of the sentence “eliminated fetal effects” or provide more info which effects. Instead of “potential causative agent” rather put “among key factors in the pathogenesis of preE”
Response: We have made the suggested change.
Critiques: In line 86, add sentence “In a rat model of the syndrome, MBG induces hypertension, proteinuria, intrauterine growth restriction, and increased weight gain, but these symptoms are prevented by an MBG antagonist.( · DOI: 10.1016/j.bbadis.2010.02.005 ),”
Response: We have added the sentence and reference.
Critiques: Lines 170-171 should be transferred elsewhere e.g. in Disscussion: “Vascular endothelial damage is part of the pathology seen in preE, and MBG promotes endothelial dysfunction in part by inhibition of proliferation as described previously by Uddin et al [22, 23].” Instead, provide here the procedure for cultivation of HUVECs, and provide reference for 3H-thymidine proliferation assay. Also, add the reference that describes that pre-treatment protocol for 1h anti-MBG humAbs is sufficient. Provide info on how you selected MBG (2 nM) concentration for this assay
Response: We have addressed these concerns.
Critiques: Next, describe the cultivation of CTB cells.Are Human CTBs primary cells used or cell line?
Response: We have added a reference to the CTB cell system and have noted the nature of and origin of the cell line.
Critiques: In line 179 the Authors mention they collected culture media for analysis of pro-angiogenic and anti-angiogenic factors. Add term “secreted protein” pro-angiogenic or angiogenic factors.
Response: We have made the suggested change.
Critiques: Line 185 “Some cells were pretreated”, describe in more details which cells.
Response: We have revised the sentence to clearly indicate that we are referring to one of the experimental groups.
Critiques: Provide more details for migration and invasion assays. Were the cells serum-starved, at what time point the scratch was made and how long did you incubate the cells after scratching, did you use chemoattractant ?
Response: We have added a reference to the assays as conducted in our laboratory.
Critiques: Line 211 add details: How did you choose concentrations for the treatment of rats with H3L2 (2.2 mg/kg/day)and anti-MBG antibody 3E9 (2.2 mg/kg/day)?
Response: We have noted our rationale for the choice of 2.2 mg/kg/day.
Critiques: Figure 2 should be corrected. In the methodology you stated 20 μg/mL concentration of MBG and in chart you wrote 20 mg/mL. Avoid the too descriptive title of Fig 2, instead the title should be “Proliferation of MBG-induced HUVEC cells and the effect of anti-MBG mAbs”. Figure legend should also be corrected, not to narratively describe the result but to provide information about details on the chart and used methodology, number of replicate experiments, compared to what you expressed the p values etc. Also, the explanation of the results in the section 258-259 is poor and lacks the description of the effects of specific tested anti-MBG mAbs.
Response: We regret the error. In Fig. 2, the concentration has been corrected to 20 μg/mL. We have edited the title and the legend, as well as the description of the results.
Critiques: Line 263 Add at the end of sentence compared to what you expressed the significant difference.
Response: We have made this change.
Critiques: Again, the description of the results of anti-MBG antibody H3L2 effects on CTB migration, proliferation and invasion are very poor, with insufficient information.
Response: We have revised the section to address these points.
Critiques: Figure 4C is of insufficient quality.
Response: We have added a new Figure
Critiques: Table 2 title is too descriptive, please correct properly. Text from 320-324 should be placed not in the title, but in the description of the results.
Response: We have revised the title and description of the results as suggested.
Critiques: Also, in table 2 add in the legend what the* stands for, the difference compared to what group? Is every other group being compared to PDS group? Description in the lines 338-341 should be in the table legend explaining these comparisons.
Response: We have made the suggested changes.
Critiques: There is no representation of baseline urine protein concentration in the Table2, and yet you describe the result in text in line 328. Please add values in the Table 2.
Response: We regret this error. There is no baseline urine protein measurement. We have noted this in our revised manuscript.
Critiques: Line 347 rephrase “as a causative agent” since it is not a single causative factor for preE, but rather state “among key factors”
Response: We have revised the wording as suggested.
Critiques: Line 356 should be a separate sentence since it refers to human studies, while line 355 refers to animal models of preE.
Response: We have made the suggested change.
Critiques: Lines 359 to 362 should be removed since they describe the phenomena already mentioned at the beginning of the same paragraph.
Response: We have deleted portions of the section that are repetitious.
Critiques: Line 373” It is known to prevent the development of hypertension, proteinuria, and intrauterine growth restriction (IUGR) in early pregnancy” Provide reference and explanation is this animal study, are you referring to reference 49?
Response: We have added the appropriate reference.
Critiques: In line 380 at the end of sentence you state “normalizes blood pressure in preE patients” and in the same sentence you mention rat model. Is this a human study or animal study? Provide reference for this statement.
Response: We have added the appropriate reference.
Critiques: Lines 384-389 provide references.
Response: We have added references.
Critiques: In lines 399 to 400 the authors state : “We note that the time of administration in the DOCA-saline model corresponds to second trimester of human gestation, illustrating a shortcoming of rodent models of preE.” Provide more discussion on this mechanisms that you represented in the illustration no 5.
Response: We have added a reference that supports the statement. It has been taken care of.
Critiques: Also, at the end of the Discussion provide more description on the limitations of the study.
Response: It has been taken care of.
Critiques: In Conclusion you state that anti-MBG antibody binds to MBG, neutralizing it,and preventing downstream signaling. But in this work, you did not analyze signaling pathways, but functional responses of CTB cells and secretory angiogenic response of HUVEC cells. Rephrase the first sentence of Conclusions and mention these actually observed results.
Response: It has been taken care of.
Critiques: Lines 413-417 should be removed from the conclusion, since this was not evaluated in this paper. You can state this at the end of Discussion as a sort of future prospects.
Response: We have moved the statement to Discussion.
Reviewer 2 Report
Comments and Suggestions for Authors
I would like to express my gratitude for being invited to review the manuscript "Neutralization of Marinobufagenin Demonstrates Efficacy In Vitro and In Vivo in Models of Preeclampsia." This is a very well-written manuscript, and I congratulate the authors on their excellent work.
I have only a few minor suggestions to enhance the overall quality of the paper. I recommend that the authors consider incorporating the following changes:
- Line 54: Please include ‘Hypertensive disorders of pregnancy (including pre-eclampsia) are the second most common cause’ and update the reference. This WHO reference is outdated, back in 2004. This is a new ref to consider: https://www.nature.com/articles/s41572-023-00417-6#:~:text=Hypertensive%20disorders%20of%20pregnancy%20(including,deaths%20per%20year22%2C23.
- Line 60: I suggest rewriting the statement, “unmet need,” which is usually a term used for family planning.
- Line 117: You can provide a reference of your previous work if required: https://www.ajog.org/action/showPdf?pii=S0002-9378%2817%2931490-4
- Line 215: what about diastolic? I suggest mentioning “BP was measured” to avoid confusion
- Line 225: Please clearly mention whether you conducted a one-way or two-way ANOVA. Also, mention the groups and factors that were considered. What software was used for analysis?
- Figure 1 appears rather hazy; I recommend including a clearer version and providing the full meanings of the acronym in the legend.
- Figure 2: Mention p<0.001 and provide the full meanings of the acronym in the legend.
- Lines 474, 503, 553, 555: These references are very old; would you consider replacing them with new ones?
- Line 482: Year of publication is absent
- Line 556: Year of publication is not complete
Author Response
Summary of revisions to manuscript: biomedicines-3480491
We would like to express our appreciation to both reviewers for their insightful comments and suggestions, which improve both the premise, conclusions, and presentation of our study. A summary of our revisions follows, organized by our response to each critique.
Reviewer 2
Critiques: Line 54: Please include ‘Hypertensive disorders of pregnancy (including pre-eclampsia) are the second most common cause’ and update the reference. This WHO reference is outdated, back in 2004. This is a new ref to consider: https://www.nature.com/articles/s41572-023-00417-6#:~:text=Hypertensive%20disorders%20of%20pregnancy%20(including,deaths%20per%20year22%2C23.
Response: We have made the suggested change.
Critiques: Line 60: I suggest rewriting the statement, “unmet need,” which is usually a term used for family planning.
Response: We have deleted the sentence, as the preceding sentences in the paragraph make it clear that preE is a major unmet need in women’s health.
Critiques: Line 117: You can provide a reference of your previous work if required: https://www.ajog.org/action/showPdf?pii=S0002-9378%2817%2931490-4
We do not think this is needed.
Critiques: Line 215: what about diastolic? I suggest mentioning “BP was measured” to avoid confusion
Response: Thank you for this suggestion. We have made the change.
Critiques: Line 225: Please clearly mention whether you conducted a one-way or two-way ANOVA. Also, mention the groups and factors that were considered. What software was used for analysis?
Response: Two-way ANOVA was conducted, and the SAS software was used.
Critiques: Figure 1 appears rather hazy; I recommend including a clearer version and providing the full meanings of the acronym in the legend.
Response: We have added the full meaning of the acronym, HUVEC. Unfortunately, no clearer version of the figure is available.
Critiques: Figure 2: Mention p<0.001 and provide the full meanings of the acronym in the legend.
Response: We have made this change.
Critiques: Lines 474, 503, 553, 555: These references are very old; would you consider replacing them with new ones?
Response: The references we have used in each case are intended to be the earliest ones of which we are aware.
Critiques: Line 482: Year of publication is absent
Response: We have added the year of publication.
Critiques: Line 556: Year of publication is not complete
Response: We have corrected this omission.
Round 2
Reviewer 1 Report
Comments and Suggestions for Authors
In line 173 provide here the procedure for cultivation of HUVECs and info on the cell line used. I already commented on this in previous report, but information is still missing.
Line 193 add manufacturer of ELISA assays and provide specification on which kits have been used. Are those provided later in line 209? If so, put them here in line 193 instead, as they are first mentioned in line 193.
Line 196 return part of the sentence that you deleted “cells were”
In is not enough to add just reference to the methodology, please provide more details for migration and invasion assays. Were the cells serum-starved, at what time point the scratch was made and how long did you incubate the cells after scratching, did you use chemoattractant. It is important for interpretation of results.
Line 227: To the previous question “How did you choose concentrations for the treatment of rats with H3L2 (2.2 mg/kg/day)and anti-MBG antibody 3E9 (2.2 mg/kg/day)?”The Authors stated “The doses of antibodies were determined to be sufficient to neutralize anticipated levels of plasma MBG.” How did you determine this? Add details in the manuscript.
Line 276 add details on which Ab showed the most profound effects on reducing MBG inhibition.
In Fig 3 check the concentration of MBG with 4.0 mg/mL H3L2, it seems it should be 4 ug/mL. Also, In Fig 4 the info on the concentration of H3L2 is missing
Poor resolution of Fig 4C. You have added the new Fig 4D, but my previous remark was on poor quality of Fig 4C not Fig 4D
Table 1 title should be rephrased “anti-MBG effects in MBG-induced preeclampsia rat model.”
The Authors haven’t addressed my previous request to provide more discussion on this mechanisms that you represented in the illustration no 5. It is necessary to address this or remove the illustration no 5.
Line 441 The Authors state “By analogy to other approved therapeutic antibodies, this antibody will likely have a plasma half-life in humans of approximately 3 weeks.” Provide more explanation and reference on this matter.
Author Response
Summary of revisions to manuscript: biomedicines-3480491
We would like to express our appreciation to the reviewer for second-round review and comments that can improve the presentation of our study. A summary of our revisions follows, organized by our response to each critique.
Comment 1. In line 173 provide here the procedure for cultivation of HUVECs and info on the cell line used. I already commented on this in previous report, but information is still missing.
Response: We have added the requested details.
Comment 2. Line 227: To the previous question “How did you choose concentrations for the treatment of rats with H3L2 (2.2 mg/kg/day)and anti-MBG antibody 3E9 (2.2 mg/kg/day)?”The Authors stated “The doses of antibodies were determined to be sufficient to neutralize anticipated levels of plasma MBG.” How did you determine this? Add details in the manuscript.
Response: We have repeated the calculation, showing that there is a 1000x excess of antibody over normal MBG levels. While MBG increases in the saline-induced model, it won't increase by 1000x. Thus, there is a very large stoichiometric excess of antibody that is more than sufficient to bind essentially all of the plasma MBG.
Comment 3. Line 276 add details on which Ab showed the most profound effects on reducing MBG inhibition.
Response: We added a sentence. In the earlier manuscript, we said the humanized antibodies also inhibited, but this is not shown in the figure, so we have deleted it.
Comment 4. In Fig 3 check the concentration of MBG with 4.0 mg/mL H3L2, it seems it should be 4 ug/mL. Also, In Fig 4 the info on the concentration of H3L2 is also 4 ug/mL.
Response: Fig. 3A, B, and C have been revised to 4 ug/mL.
​ Comment 5. Poor resolution of Fig 4C. You have added the new Fig 4D, but my previous remark was on poor quality of Fig 4C not Fig 4D
​ Response: We have improved the quality of Fig. 4C.
Comment 6. Table 1 title should be rephrased “anti-MBG effects in MBG-induced preeclampsia rat model.”
​ Response: The reviewer is referring to Table 2. We changed the title as requested.
Comment 7. Line 441 The Authors state “By analogy to other approved therapeutic antibodies, this antibody will likely have a plasma half-life in humans of approximately 3 weeks.” Provide more explanation and reference on this matter.
​ Response: We have added a reference.
Comment 8. Line 193 add manufacturer of ELISA assays and provide specification on which kits have been used. Are those provided later in line 209? If so, put them here in line 193 instead, as they are first mentioned in line 193. Line 196 return part of the sentence that you deleted “cells were”
In is not enough to add just reference to the methodology, please provide more details for migration and invasion assays. Were the cells serum-starved, at what time point the scratch was made and how long did you incubate the cells after scratching, did you use chemoattractant. It is important for interpretation of results.
Response: We have rewritten the migration and invasion assay sections.
Comment 9. The Authors haven’t addressed my previous request to provide more discussion on this mechanisms that you represented in the illustration no 5. It is necessary to address this or remove the illustration no 5.
Response: We have rewritten the migration and invasion assay section.